# Autophagy a Close Relative of AML Biology

**DOI:** 10.3390/biology10060552

**Published:** 2021-06-18

**Authors:** Carine Joffre, Charlotte Ducau, Laura Poillet-Perez, Charly Courdy, Véronique Mansat-De Mas

**Affiliations:** 1Centre de Recherches en Cancérologie de Toulouse (CRCT), Université de Toulouse, Inserm, CNRS, 31037 Toulouse, France; charlotte.ducau@inserm.fr (C.D.); laura.poillet-perez@inserm.fr (L.P.-P.); charly.courdy@inserm.fr (C.C.); 2Centre Hospitalo-Universitaire (CHU) de Toulouse, Institut Universitaire du Cancer de Toulouse-Oncopole (IUCT-O), 31059 Toulouse, France

**Keywords:** autophagy, mitophagy, hematopoiesis, AML, therapy

## Abstract

**Simple Summary:**

Acute myeloid leukemia (AML) is the most common acute leukemia in adults. Despite a high rate of complete remission following conventional chemotherapy, the prognosis remains poor due to frequent relapses caused by relapse-initiating leukemic cells (RICs), which are resistant to chemotherapies. While the development of new targeted therapies holds great promise (e.g., molecules targeting IDH1/2, FLT3, BCL2), relapses still occur. Therefore, a paramount issue in the elimination of RICs is to decipher the AML resistance mechanisms. Thus, it has been recently shown that AML cells exhibit metabolic changes in response to chemotherapy or targeted therapies. Autophagy is a major regulator of cell metabolism, involved in maintaining cancer state, metastasis, and resistance to anticancer therapy. However, whether autophagy acts as a tumor suppressor or promoter in AML is still a matter of debate. Therefore, depending on molecular AML subtypes or treatments used, a better understanding of the role of autophagy is needed to determine whether its modulation could result in a clinical benefit.

**Abstract:**

Autophagy, which literally means “eat yourself”, is more than just a lysosomal degradation pathway. It is a well-known regulator of cellular metabolism and a mechanism implicated in tumor initiation/progression and therapeutic resistance in many cancers. However, whether autophagy acts as a tumor suppressor or promoter is still a matter of debate. In acute myeloid leukemia (AML), it is now proven that autophagy supports cell proliferation in vitro and leukemic progression in vivo. Mitophagy, the specific degradation of mitochondria through autophagy, was recently shown to be required for leukemic stem cell functions and survival, highlighting the prominent role of this selective autophagy in leukemia initiation and progression. Moreover, autophagy in AML sustains fatty acid oxidation through lipophagy to support mitochondrial oxidative phosphorylation (OxPHOS), a hallmark of chemotherapy-resistant cells. Nevertheless, in the context of therapy, in AML, as well as in other cancers, autophagy could be either cytoprotective or cytotoxic, depending on the drugs used. This review summarizes the recent findings that mechanistically show how autophagy favors leukemic transformation of normal hematopoietic stem cells, as well as AML progression and also recapitulates its ambivalent role in resistance to chemotherapies and targeted therapies.

## 1. Introduction

Acute myeloid leukemia (AML) arises from malignant clonal expansion of undifferentiated myeloid precursors, leading to bone marrow failure. Despite response to conventional induction therapy in many patients, AML is characterized by a high risk of relapse and poor prognosis [1]. Recently, extensive molecular characterization studies have led to a deeper understanding of the complex AML heterogeneity and to the development of targeted therapies [2]. Besides the improvement in the genetic knowledge of AML physiopathology, several important cellular mechanisms such as autophagy have been identified to be deregulated in AML and involved in its development and therapeutic resistance [3,4].

Macroautophagy (referred to here as autophagy) is a catabolic process that degrades and recycles intracellular components, such as damaged organelles and macromolecules, as an adaptative survival mechanism activated during cellular stresses. In this process, a double-membrane structure called the phagophore matures into the autophagosome, which then fuses with the lysosome to form the autolysosome, where the enzymatic degradation of the intravesicular content happens. Several key proteins are involved in the autophagic process: for example, ULK1 (Unc-51 like autophagy activating kinase 1), FIP200/RB1CC1 (focal adhesion kinase family interacting protein of 200 kD/RB1 inducible coiled-coil 1), BECLIN1, VPS34, and ATG14 are involved in the phagophore formation, while ATG5, ATG7, ATG12, and LC3-II (microtubule-associated protein 1 light chain 3) regulate the autophagosome formation [5]. In addition to bulk degradation, autophagy can trigger the selective elimination of impaired or extra organelles, such as mitochondria, the powerhouse of the cell, recently shown to be a key organelle in AML therapeutic resistance [6,7]. This selective autophagy, called mitophagy, allows the specific clearance of damaged, dysfunctional, or superfluous mitochondria, contributing to the maintenance of a functional pool of mitochondria. The most studied mitophagy pathway is mediated by PINK1 (PTEN-induced putative kinase 1), a serine/threonine kinase, and PARKIN, an E3 ubiquitin ligase [8] that promotes the recruitment of damaged mitochondria to autophagy adaptors, such as NDP52 and OPTINEURIN. This then triggers the recruitment of LC3 to the mitochondria through its LC3 interacting region (LIR) inducing engulfment of the mitochondria in the autophagosome and its degradation [9]. However, pathways independent of PINK1/PARKIN are now described and require other mitophagy receptors such as BNIP3 (BCL2/adenovirus E1B 19 kDa protein-interacting protein 3), NIX/BNIP3L, FUNDC1 (FUN14 domain-containing protein 1) and BCL2L13 (BCL2 like 13) [10].

In agreement with numerous studies, this review illustrates how autophagy’s participation in AML development, as well as therapeutic resistance, transcends beyond its original function as merely a recycling process.

## 2. The Role of Autophagy: From Normal Hematopoiesis to Leukemia

### 2.1. Autophagy in Normal Hematopoietic Stem Cell Maintenance and Function

The precise role of autophagy in the regulation of stem cells and progenitor cells is not completely elucidated. However, during the last few years, several works, especially on murine models, have shown that intrinsic autophagy is required for the maintenance and function of normal hematopoietic stem cells (HSC).

Indeed, deletion of autophagy-related genes like *FIP200/RB1CC1* and *ATG7* in mice reduces fetal and adult HSC frequencies, respectively, induces loss of reconstituting capacity, and may impair viability. Moreover, *FIP200/RB1CC1* or *ATG7* deficiency in the hematopoietic system results in concomitant mitochondria and reactive oxygen species (ROS) accumulation and DNA damage [11,12]. Mice deficient for *FIP200/RB1CC1* also displayed perinatal lethality associated with anemia and increased frequency of immature erythroid cells [11].

In another murine model, deletion of *ATG12* decreased HSC self-renewal and function and led to an increase in mitochondrial mass and respiration. The authors also showed that autophagy actively degrades active and healthy mitochondria in order to maintain quiescence and stemness, and therefore becomes increasingly necessary with age to preserve the functions of old HSCs [13].

A human model has also shown that immature CD34 positive cord blood cells have higher autophagic flux than CD34 negative differentiated cells. *ATG5* or *ATG7* depletion resulted in a decrease in HSC frequency and to a strong reduction in cell expansion upon myeloid/erythroid differentiation due to reduced cell cycle progression and increased apoptosis [14].

Accumulation of mitochondria observed in these autophagy-deficient models may be due to the impairment of mitophagy. Mitophagy plays a central role in cell metabolism and redox balance and consequently in HSC maintenance and functions (Figure 1).

Interestingly, mitophagy is necessary for long-term HSC (LT-HSC) [15]. Indeed, mitophagy helps to maintain a healthy and limited network of mitochondria in LT-HSC, so ROS levels stay low and thus prevent their differentiation in short-term HSC (ST-HSC), characterized by reduced self-renewal capacity [13,16,17]. Therefore, autophagy-deficient HSCs show a reduction in active mitochondria clearance and then leave their quiescent state, resulting in an increased myeloid differentiation. Ito and colleagues showed that this mitochondrial clearance is achieved through induction of PINK1/PARKIN-dependent mitophagy in a purified Tie2 positive HSC population [18]. PARKIN-mediated mitophagy has also been involved in a more global HSC population [17], where an increase in mitochondrial mass and metabolism drives HSC commitment into differentiation.

### 2.2. Autophagy in Leukemia Initiation and Development

In addition to HSC maintenance, functional autophagy seems to be essential in AML initiation and development.

As an example, in Vav*-ATG7−/−* mice, loss of autophagy in the hematopoietic compartment causes an increased aberrant myeloid expansion and death of the mice within weeks. Histopathological analyses revealed severe myeloid dysplasia and infiltrating myeloid blast cells evoking AML [12]. Impairment of autophagy in the same model activates the NOTCH signaling pathway, blocking HSC differentiation. We can speculate that this phenomenon could induce a switch from normal to preleukemic HSC, as low autophagy levels and upregulated NOTCH have been observed in human leukemia patient’s samples [19].

Functional autophagy also seems to be essential to avoid the evolution of myelodysplastic syndrome (MDS) (“pre-leukemic state”) to AML. Actually, a mutation in the splicing factor U2AF35 found in MDS leads to abnormal *ATG7* pre-mRNA processing due to altered splicing, and consequently decreased ATG7 and autophagy levels. This autophagy defect predisposes cells to secondary mutations resulting in malignant transformation [20].

Moreover, whole-exome sequencing analysis of myeloid neoplasms has shown that mutations of autophagy-related genes are more prevalent in MDS and AML, co-occurred with unfavorable mutations, and are associated with inferior survival [21]. This observation suggests that loss of genetic autophagy regulation may be a cooperative mechanism leading to leukemogenesis. Consistent with these data, as an example, heterozygous loss of *ATG5* enhances disease progression and aggressiveness of an MLL-ENL (mixed lineage leukemia—eleven nineteen leukemia)-driven AML mouse model [22].

Given the importance of mitophagy in regulating cell metabolism and HSC functions, it is not surprising that damages in mitochondrial dynamics could play a role in leukemogenesis and/or AML development (Figure 1). Therefore, loss of autophagy in HSC, accompanied by an increase in mitochondrial mass, has been implicated in myeloproliferative disorders and tissue infiltration by myeloid cells [12,13].

Altogether, these studies suggest that maintaining autophagy flux within the immature HSC compartment is essential for their survival. Autophagy impairment may lead to mitochondria and ROS accumulation, DNA damage, and increased proliferation, thus promoting preleukemic phenotype. The critical role of autophagy in hematopoiesis as well as the dysregulation of autophagy genes in human AML, strongly implicates its role in leukemogenesis (Figure 1).

### 2.3. Contribution of Autophagy to AML Metabolism during Leukemic Development

Even though autophagy is described as a major regulator of cellular metabolism, only a few studies focused on its role in AML energetics. Recent work from Bosc et al. indicates that lipophagy, the specific degradation of lipid droplets by autophagy, supplies free fatty acids to maintain mitochondria energy metabolism [23]. In this context, the authors demonstrated that, in return, mitochondrial respiratory chain activity positively regulates autophagy. Moreover, this interplay between mitochondria and the autophagy mechanism occurs through the formation of contact sites between the mitochondria and the endoplasmic reticulum in AML cells. These inter-organelle regions called MERCs for mitochondria endoplasmic reticulum contact sites were described as a subcellular localization for autophagosomes formation in different cell types [24,25]. Therefore, the mitochondria-autophagy cross-talk allows the fine regulation of OxPHOS described as being a prerequisite for therapy resistance in AML [6].

Another means for autophagy to regulate OxPHOS metabolism is through mitochondria turnover regulation *via* mitophagy (Figure 1). Accordingly, Pei and colleagues showed that FIS1 (Fission 1), a protein involved in mitophagy, is upregulated in an AMPK-dependent manner in leukemic stem cells (LSC), known to drive AML development [26]. Loss of *FIS1* disrupts mitophagy activation, leading to impaired maintenance of the LSC population, whereas it has very little impact on healthy HSC. Moreover, high expression of FIS1 protein has been associated with poor prognosis in AML patients [27], suggesting a potential role for AMPK/FIS1 axis-targeting in AML treatment. Another protein shown to induce poor overall survival when overexpressed is P62/SQSTM1 (sequestosome 1), a selective autophagy receptor. Indeed, P62 protein is necessary for the development and progression of AML in vivo through the induction of PINK1/PARKIN-independent mitophagy [28]. Interestingly, *P62* deletion led to the accumulation of damaged mitochondria and ROS in AML cells but had no severe impact on normal HSC, thus having no effect on hematopoiesis. Another very recent study demonstrated the major role of mitophagy during AML development and progression. In fact, Li et al. showed that the new P62 inhibitor, XRK3F2, exerts its antileukemic activity *via* impairing mitophagy in leukemia-initiating cells (LICs), characterized by a decreased colocalization between LC3 and mitochondrial proteins concomitant with an accumulation of dysfunctional mitochondria [29]. Similar to P62 depletion [28], XRK3F2 does not induce excessive apoptosis in normal hematopoietic stem cells, indicating that this compound selectively targets primary AML progenitors.

In summary, normal hematopoiesis and leukemic hematopoiesis share several common characteristics, with autophagy being essential to both of them. Autophagy is indeed required for maintenance and function of normal HSC [13] but also of LSC [26] mainly by regulating mitochondrial mass and hence mitochondrial functions highlighting, once again, the importance of mitophagy in leukemogenesis.

## 3. Recurrent Genetic AML Alterations Associated with Autophagy

AML has been recently characterized into specific molecular subtypes leading to a more accurate classification [2]. Several studies have shown that, depending on the molecular subtype studied, autophagy may promote protumoral or antitumoral effect (Figure 2).

FLT3 (FMS-like tyrosine kinase 3) is a receptor tyrosine kinase commonly mutated in AML patients. The most frequent mutations in this gene occur *via* internal tandem duplication (*FLT3-*ITD) in the juxtamembrane domain or through point mutations, usually involving the tyrosine kinase domain (*FLT3-*TKD). Both mutations result in FLT3 constitutive activation, leukemic cell proliferation, and survival. *FLT3*-ITD confers poor prognosis outcomes in AML patients [30]. Interestingly, FLT3-ITD activity has been shown to increase basal autophagy in AML cells, which is required for AML cell proliferation in vitro and for leukemic cell survival in a xenograft mouse model. ATF4 (activating transcription factor 4) was identified as an essential mediator of FLT3-ITD-induced autophagy. Moreover, downregulation of autophagy or *ATF4* inhibited AML cell proliferation and improved mice survival. In addition, autophagy inhibition in FLT3-TKD cells, which are resistant to the FLT3 inhibitor quizartinib (AC220), is also able to inhibit proliferation both in vitro and in vivo [31].

Autophagy not only contributes to proliferation downstream of FLT3-ITD receptor but may also be involved in its degradation. As an example, it has been shown that RET, a tyrosine kinase receptor frequently activated in AML, mediates autophagy suppression in a mTORC1 (mammalian target of rapamycin complex1)-dependent way, leading to mutant FLT3 receptor stabilization. These results suggest that RET inhibition can restore autophagy and FLT3 degradation and therefore may represent an interesting therapeutic approach in this model [32].

Activating mutations of the KIT receptor tyrosine kinase are also frequently detected in core-binding factor AML and are associated with a higher risk of relapse [33]. KITD816V activity is associated with increased basal autophagy level in a STAT3-dependent manner, contributing to AML cell proliferation and cell survival in vitro. Moreover, in the same study, the authors showed that deletion of the key autophagy gene *ATG12* strongly reduced tumor burden and improved survival of NSG-engrafted mice [34].

These studies suggest that targeting ATF4 or autophagy in *FLT3*-mutated AML patients and autophagy or STAT3 in KITD816V AML patients may represent promising therapeutic strategies.

*NPM1* (nucleophosmin 1) mutations are the most frequent genetic alteration in AML and promote aberrant cytoplasmic localization of NPM1 protein [35]. Several publications have shown that autophagy is activated in NPM1-mutated AML cells and contributes to leukemic cell survival [36,37]. Mechanistically, mutant NPM1 can interact with the tumor suppressor protein PML (promyelocytic leukemia protein) and mediates its relocalization in the cytoplasm as well as stabilization, then promoting autophagy *via* AKT signal [37]. Another group demonstrated that in NPM1-mutated AML, the glycolytic enzyme PKM2 (pyruvate kinase M2) activates autophagy by increasing the phosphorylation of key autophagy protein BECLIN1 and contributes to cell survival [36]. Finally, NPM1 mutant can also interact with ULK1, a core autophagy protein that promotes TRAF6-dependent ULK1 ubiquitination *via* miR-146, maintaining ULK1 stability and activity and promoting autophagic cell survival [38].

*TP53* mutations are frequently associated with complex karyotypes and therapy-related AML. The link between autophagy and AML depends on *TP53* status. For example, in AML *TP53* wild-type (*TP53*WT) but not in *TP53*-mutated AML, inhibition of autophagy by *ATG5* or *ATG7* knockdown increased P53 expression and triggered a BAX and PUMA-dependent apoptosis response [39]. This observation suggests that intact P53 function is needed to exert its tumor suppressor activity and that inhibition of autophagy represents a therapeutic strategy in *TP53* WT AML but not in *TP53* mutated AML.

Besides these observations, the association between autophagy and other genetic recurrent alterations, such as *IDH* (isocitrate dehydrogenase) or *DNMT3α* (DNA methyltransferase 3α) mutations, are suggested in some cancers but need further studies in AML [40,41]. Moreover, autophagy activation upon treatment may also contribute to oncoproteins degradation like PML-RARα (promyelocytic leukemia/retinoic acid receptor α), KMT2α (lysine methyltransferase 2α) fusion proteins or FLT3-ITD, mutant *TP53* (see below “autophagic response upon therapy”).

## 4. Autophagic Response upon Therapy in AML

As with other types of cancer, AML is no exception to the rule and numerous small molecules/drugs can induce autophagy, which can be either cytotoxic or cytoprotective (Figure 2).

### 4.1. Therapy-Induced Cytotoxic Autophagy

Depending on the drug used, the induced cytotoxic autophagy leads to cell death by different means. One of them is the specific degradation of oncogenes or fusion proteins, such as PML-RARα, required for AML cell proliferation and survival by autophagy. Upon all-trans retinoic acid (ATRA) treatment, PML-RARα is degraded through autophagy [42]. This differentiation agent is used to treat the acute promyelocytic leukemia subtype and induces autophagy through mTORC1 inhibition. In accordance, rapamycin, a mTORC1 inhibitor, similar to ATRA, leads to PML-RARα degradation *via* autophagy [43]. The autophagy core machinery implicated in PML-RARα degradation can be either the autophagy adaptor P62 [43] or the protein ALFY that acts as a molecular scaffold between PML-RARα and the autophagy process [44]. The mutant receptor FLT3-ITD, was also shown to be specifically degraded *via* autophagy when AML cells were treated with the proteasome inhibitor, Bortezomib, leading to apoptosis in vitro and leukemia development inhibition in vivo [45]. Another study indicates that arsenic trioxide also induces FLT3-ITD degradation through autophagy [46]. These studies imply that oncogenes or fusion proteins elimination through the autophagy pathway may offer therapeutic opportunities. For this purpose, researchers identified the HSP90 (Heat shock protein 90 kDa) inhibitor, 17-AAG, as an efficient molecule to induce autophagy and degradation of mutated P53 that participates in leukemia progression [47].

Autophagy-induced cell death upon treatment could also be a consequence of the exacerbated metabolism displayed by leukemic cells. In AML cells, mTORC1 is constitutively activated, and targeting this pathway could be a therapeutic opportunity. Bouscary and Tamburini’s laboratory showed that inhibiting mTORC1 either pharmacologically [48] or by preventing its activation through glutamine removal [49] induces activation of the autophagic flux. Unexpectedly, this induced autophagy is implicated in either AML cell death or survival. Indeed, a low concentration of the mTORC1 inhibitor AZD8055 leads to the induction of cytotoxic autophagy, whereas a high concentration or the use of L-asparaginase that harbors glutaminase activity and inhibits mTORC1 promotes cytoprotective autophagy. Further studies are needed to understand mechanistically how a strong induction of autophagy could lead to the opposite effect on cell survival depending on the concentration of the drug used. This apparent discrepancy also occurs following FLT3 inhibitor treatment. Indeed, FLT3-ITD inactivation with several FLT3 inhibitors (including AC220) used at high concentration (μM range) induced a lethal mitophagy due to ceramides accumulation in the mitochondria [50], whereas their use at a lower concentration (nM range) inhibited FLT3-ITD-induced autophagy [31] (see above “Recurrent genetic AML alterations associated with autophagy” paragraph). In the first study [50], pharmacological inhibition of the FLT3-ITD receptor reduces PKA activity, thus leading to CerS1 (ceramide synthase 1) translocation to the mitochondrial membrane. At this specific location, CerS1 produces C18-ceramides that directly interact with LC3B-II, inducing mitophagy that finally leads to cell death. Treatment of FLT3-ITD AML-engrafted mice with a C18-ceramide analogue induced lethal mitophagy in blasts. In contrast, in the second work [31], the FLT3 inhibitor AC220 (concentration of 3 nM) reduced autophagy through ATF4 expression inhibition. Thus, it would be of interest to study how other FLT3 inhibitors (midostaurin and gilteritinib) used in clinics in combination with standard chemotherapies regulate autophagy in AML patients harboring FLT3-ITD mutations.

Recently, mitophagy has also been involved in AML cell death upon association of C6-Ceramide to tamoxifen in FLT3-ITD positive and FLT3-ITD negative cell lines. Mechanistically, this induced lethal mitophagy occurred *via* mTOR inhibition [51]. Hence, the ceramide/mitophagy pathway might be considered as a target for treatment in AML.

Recently, a mammalian cholesterol metabolite, dendrogenin A (DDA), was shown to potentiate the effect of cytarabine (AraC), the front-line chemotherapy used in clinics. This metabolite leads to cell death by inhibiting a sterol isomerase, therefore perturbing cholesterol metabolism and inducing lethal autophagy [52]. Mechanistically, DDA triggers autophagy *via* the activation of the nuclear receptor LXRβ (liver-X-receptorβ) but also *via* its concomitant action on cholesterol metabolism [53]. By triggering lethal autophagy, DDA also potentiates the antileukemic activity of anthracyclines (idarubicin or daunorubicin, DNR), which represents, in association with AraC, the gold standard AML therapy [54].

### 4.2. Autophagy and Therapy Resistance in AML

Even though data are sparse and incomplete, studies suggest that AraC could increase autophagy in vitro [55]. Accordingly, simultaneously disrupting LIR on several autophagic receptors required for selective autophagy sensitizes AML cells to AraC treatment [56]. Moreover, this therapy-induced autophagy was described as cytoprotective, since its inhibition potentiates the therapeutic effect of AraC in vivo in MLL-ENL murine model [57]. Indeed, the combination of AraC and *ATG7* depletion in this particular mouse model results in increased apoptosis in peripheral blood leukemia cells, in a reduced number and frequency of LICs, and enhanced overall survival compared to AraC treated mice alone. Similar results were obtained in NSG-engrafted mice with OCI-AML3 depleted for *ATG7* [58]. Interestingly, the authors also showed that, in co-culture systems, the concomitant depletion of *ATG7* in AML and stromal cells increased AraC and idarubicin-induced apoptosis. This study is the first to report the potential contribution of autophagy occurring in a tumor microenvironment (host autophagy) to chemoresistance. The bone marrow microenvironment, which is constituted of numerous cell types such as mesenchymal stem cells (MSC), adipocytes, and endothelial cells, has long been known to support leukemia cell survival and chemotherapy resistance [59]. One explanation among many is the hypoxic environment of the bone marrow, a known inducer of chemoresistance in AML [60]. This hypothesis is supported by a recent study showing that hypoxia induces cytoprotective autophagy. In fact, pharmacological inhibition of autophagy reduced the hypoxia-induced chemoprotection in vitro, and its combination with AraC inhibits AML growth in vivo [61]. The authors also indicated that autophagy inhibition triggers an accumulation of damaged mitochondria and highlights the importance of mitophagy in therapeutic resistance.

Additional studies are needed, especially in vivo, with the deletion of other Atg genes than *ATG7* in order to evaluate the genuine contribution of autophagy to AraC resistance. Few studies have also investigated the contribution of the autophagy process upon DNR response, an anthracycline commonly used in combination with AraC. As with AraC, DNR is described to trigger cytoprotective autophagy mediated by the ULK1-AMPK pathway and independently of mTORC1. Thus, a selective kinase inhibitor of ULK1 (SBI-0206965) enhanced DNR cytotoxicity in vitro [62].

Another frequently used drug in AML patients is the hypomethylating agent azacytidine (Aza). Recently, Aza treatment has also been shown to induce autophagy [63]. The autophagy inhibitor ROC-325, a chloroquine derivative, potentiates the antileukemic activity of Aza in vitro and in vivo, suggesting a cytoprotective role of Aza-induced autophagy.

Targeted therapies are also able to induce autophagy in AML. A complete and recent review nicely covers this specific point [4]. In t(8;21) AML cells expressing the fusion oncoprotein AML1–ETO (eight-twenty-one), targeted therapies such as histone deacetylase inhibitors (HDACis) are warranted. In 2013, it was shown that HDACis induced prosurvival autophagy to limit drug-induced cell death [64]. In this AML subtype, the treatment-induced autophagy, in contrast with PML-RARα, did not lead to AML1-ETO degradation but rather to cytoprotective autophagy. Therefore, inhibition of autophagy in combination with HDACi could be useful to improve treatment efficiency.

Other FDA-approved targeted therapies (IDH1/2 and BCL2 inhibitors) hold great promise, however, therapeutic resistance is already being observed. While studies in other cancers (e.g., glioblastoma) suggest that autophagy inhibition could improve mutated-IDH targeted therapies [65,66], whether this is the case in AML still remain unknown. The direct link between the BCL2 inhibitor, venetoclax, and autophagy also needs to be studied in AML, even though a recent study suggests that elevated autophagy protects AML cells against venetoclax-induced apoptosis [67].

BET (bromodomain and extra terminal domain family) inhibitors usually used in AML patients overexpressing or displaying an aberrant activation of c-MYC were also shown to activate autophagy to counteract the effect of those inhibitors [68].

All these studies indicate there is a real need to understand whether autophagy is activated or not following treatment; this information will help determine whether autophagy should be inhibited or promoted in order to optimize AML therapy. If the cytoprotective role of AraC-induced autophagy is confirmed, inhibiting autophagy either during AraC treatment or during the consolidation phase in clinic could represent an interesting-based AML target strategy. Moreover, since autophagy has been shown to participate in tyrosine kinase inhibitor (TKI) resistance, a combination of TKI and autophagy inhibitor could also improve treatment, for example, in FLT3-ITD AML. Currently, chloroquine or hydroxychloroquine are the only clinically available drugs to inhibit autophagy. Unfortunately, these drugs are not specific to this process, and clinical trials show that although these molecules have very little toxicity in combination with other anticancer treatments, their ability to modulate autophagy within the tumor remains weak [69]. The development of specific autophagy inhibitors with improved bioavailability is, therefore, a major challenge for the pharmaceutical industry. However, autophagy inhibition is not a good strategy for all types of AML, as it has been shown for *TP53* mutated AML (see Section 3). Another interesting approach is, on the other hand, to promote autophagy to eliminate oncoproteins like FLT3-ITD or PML-RARα.

In conclusion, modulating autophagy in AML represents a promising therapeutic approach but needs to be studied in each different molecular subgroup and each treatment approach. Furthermore, in parallel, it is crucial to apprehend how leukemic and host autophagy favor or impede chemotherapies or targeted therapy resistance, since those molecular mechanisms remain poorly understood.

## 5. Conclusions

Based on numerous studies, autophagy appears to have a critical role in AML development and progression. Reduced autophagy may lead to the promotion of preleukemia state, predisposing cells to secondary mutations resulting in the leukemic transformation that often relies on elevated autophagy. Evidence for the importance of mitophagy in AML is also increasing, as LSC functions and survival seem to be dependent on the regulation of an active mitochondria pool. Thus, autophagy modulation may be an efficient approach to improve chemotherapeutic antileukemic regimens or novel targeted therapies in AML. However, as treatment-induced autophagy may be cytotoxic or cytoprotective, and AML represents a heterogeneous disease, further studies are necessary to understand the role of autophagy in each AML subtype to optimize treatments and overcome resistance. The development of novel specific autophagy modulators targeting leukemic cells could further demonstrate the therapeutic benefit of targeting autophagy in AML.

## Figures and Tables

**Figure 1 biology-10-00552-f001:**
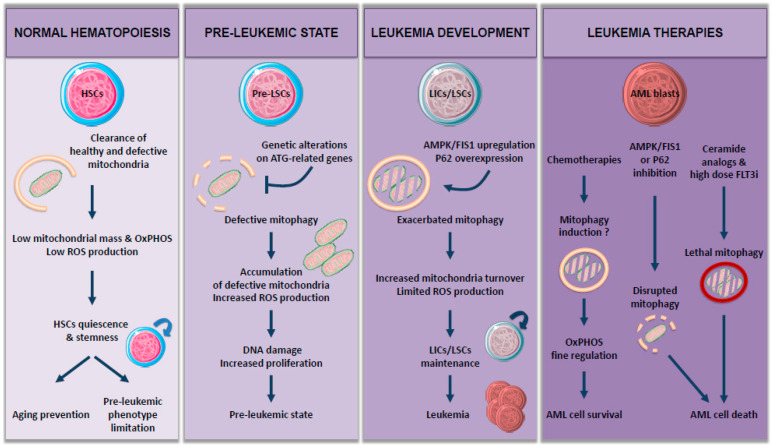
The role of mitophagy: from normal hematopoiesis to leukemia.

**Figure 2 biology-10-00552-f002:**
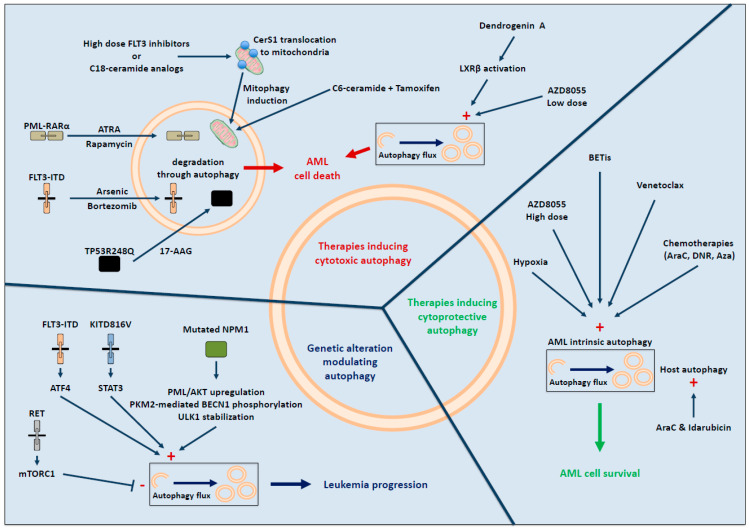
Contribution of autophagy in AML development and treatment.

## Data Availability

Not applicable.

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
