# Peer review of "Autophagy a Close Relative of AML Biology"

_biology, 2021, doi:10.3390/biology10060552_

Round 1

Reviewer 1 Report

The paper Autophagy a close relative of AML biology is a well designed study. The authors discussed new information, comprehensively described that definetly must be disseminated. But not in the current form because there are numerous mistakes on genetics terms, information without citations, papers cited on the manuscript that do not appear on the references list and the references are not according to the Biology Journal. 

The gene name shoud be written with Italic (ex ATG14). The protein name without. I suggest to make clear differentions between genes and proteins were both are disscused (ex- page 1, 41-46) EX- ATG14 gene codified the ATG protein

Tian et al 2014 do not appear on reference list, Please check all the references at cite the paper according to the Biology journal reccomandations

Were is mentioned the paper of Tian et al 2014 is written that high expression of FIS1... So you disscus about FIS1 gene. The genes name shoud be written with capitals and Italic 

On the page 4, row 161-163 the authors cited the paper of Nguyen et al., 2019 as a general disscution and then 163-167 I suppost that the conclusions of the mentioned study are writeen without references  (The protein P62 is necessary for the development and progression of AML in vivo through the induction of PINK1/PARKIN-independent mitophagy. Deletion of p62 led to the accumulation of dam aged mitochondria and ROS in AML cells but had no severe impact on normal HSC, thus having no effect on hematopoiesis ) Please mention the source and check this to the whole manuscript. 

AML has been recently characterized into specific molecular subtypes leading to a more accurate classification. Several studies have shown that, depending on the molecular subtype studied and/or the treatment used, autophagy may promote pro-tumoral or anti- tumoral effect. Please cite the paper that sustain yours information

ITD and TKD for FLT3 must be without Italic

Page 6, 183-193 several information without citations 

FLT3-ITD expression??? 

NPM1 (nucleophosmin 1) mutations are the most frequent genetic alteration in AML and promote aberrant cytoplasmic localization of NPM1 protein- NPM1 gene mutations .... and cite the source 

Activating mutations of the KIT receptor tyrosine kinase are also frequently detected in core-binding factor AML and are associated with higher risk of relapse. Similarly, KITD816V expression is associated with increased basal autophagy level in a STAT3 dependent manner, contributing to AML cell survival. Moreover, deletion of the key autophagy protein Atg12 strongly reduced tumor burden and improved survival of NSG-engrafted mice (Larrue et al., 2019). Genes with Italic, mention the paper cited

And I can continue in this way to the end of the manuscript...

I encourage to make the structural changes of the manuscript, consisting mainly on genetics terminology (a geneticist easy can do this) and correct citations and references list, and to resubmit the paper. Once that the mentioned are corrected, in my opinion the paper can be accepted for publication.

Author Response

The paper Autophagy a close relative of AML biology is a well designed study. The authors discussed new information, comprehensively described that definitely must be disseminated. But not in the current form because there are numerous mistakes on genetics terms, information without citations, papers cited on the manuscript that do not appear on the references list and the references are not according to the Biology Journal. 

We thank reviewer#1 for his/her constructive comments.

Point 1: The gene name should be written with Italic (ex ATG14). The protein name without. I suggest to make clear differentions between genes and proteins were both are discussed (ex- page 1, 41-46) EX- ATG14 gene codified the ATG protein

We apologize for not having correctly written gene and protein names throughout the manuscript. As suggested, we have now written the gene names in capitals and Italic and the protein names in capitals only.

Point 2: Tian et al 2014 do not appear on reference list, Please check all the references at cite the paper according to the Biology journal recommendations.

We have now added the missing references and adjusted them following the Biology journal recommendations (i.e. ACS. Style)

Point 3: Were is mentioned the paper of Tian et al 2014 is written that high expression of FIS1... So you discuss about FIS1 gene. The genes name should be written with capitals and Italic 

As mentioned above, we have now written the gene names in capitals and Italic and the protein names in capitals only.

Point 4: On the page 4, row 161-163 the authors cited the paper of Nguyen et al., 2019 as a general discussion and then 163-167 I suppose that the conclusions of the mentioned study are written without references  (The protein P62 is necessary for the development and progression of AML in vivo through the induction of PINK1/PARKIN-independent mitophagy. Deletion of p62 led to the accumulation of damaged mitochondria and ROS in AML cells but had no severe impact on normal HSC, thus having no effect on hematopoiesis) Please mention the source and check this to the whole manuscript. 

We apologize for not making this clear enough in the first version of the manuscript. We have now inserted the reference of Nguyen et al. 2019 line 179 and made modifications as follow: “Another protein shown to induce poor overall survival when overexpressed is P62/SQSTM1 (Sequestosome 1), a selective autophagy receptor. Indeed, P62 protein is necessary for the development and progression of AML in vivo through the induction of PINK1/PARKIN-independent mitophagy [28]. Interestingly, P62 deletion led to the accumulation of damaged mitochondria and ROS in AML cells but had no severe impact on normal HSC, thus having no effect on hematopoiesis.’

Point 5: AML has been recently characterized into specific molecular subtypes leading to a more accurate classification. Several studies have shown that, depending on the molecular subtype studied and/or the treatment used, autophagy may promote pro-tumoral or anti- tumoral effect. Please cite the paper that sustain yours information.

We are now citing the following reference: Papaemmanuil et al, NEJM 2016

Treatments have been deleted and the studies are described below.

Point 6: ITD and TKD for FLT3 must be without Italic

FLT3-ITD and FLT3-TKD are now written correctly

Point 7: Page 6, 183-193 several information without citations. 

We are now citing the following reference:

  • Daver et al, Leukemia 2019

Point 8: FLT3-ITD expression??? 

We agree with the referee and we have changed the word “expression” to the word “activity” to be more accurate.

Point 9: NPM1 (nucleophosmin 1) mutations are the most frequent genetic alteration in AML and promote aberrant cytoplasmic localization of NPM1 protein- NPM1 gene mutations .... and cite the source 

We have written the NPM1protein and NPM1 gene.

We are now citing the following reference:

  • Fallini et al, NEJM 2005

Point 10: Activating mutations of the KIT receptor tyrosine kinase are also frequently detected in core-binding factor AML and are associated with higher risk of relapse. Similarly, KITD816V expression is associated with increased basal autophagy level in a STAT3 dependent manner, contributing to AML cell survival. Moreover, deletion of the key autophagy protein Atg12 strongly reduced tumor burden and improved survival of NSG-engrafted mice (Larrue et al., 2019). Genes with Italic, mention the paper cited.

After the following sentence “Activating mutations of the KIT receptor tyrosine kinase are also frequently detected in core-binding factor AML and are associated with higher risk of relapse”,

we have added the following reference:

  • Paschka et al, J Clin Oncol 2006

We have also modified the sentence “Similarly, KITD816V expression is associated with increased basal autophagy level in a STAT3 dependent manner, contributing to AML cell survival. Moreover, deletion of the key autophagy protein Atg12 strongly reduced tumor burden and improved survival of NSG-engrafted mice (Larrue et al., 2019)” to “KITD816V activity is associated with increased basal autophagy level in a STAT3-dependent manner, contributing to AML cell proliferation and cell survival in vitro. Moreover, in the same study, authors showed that deletion of the key autophagy gene ATG12 strongly reduced tumor burden and improved survival of NSG-engrafted mice (Larrue et al., 2019).”

Point 11: And I can continue in this way to the end of the manuscript...

I encourage to make the structural changes of the manuscript, consisting mainly on genetics terminology (a geneticist easy can do this) and correct citations and references list, and to resubmit the paper. Once that the mentioned are corrected, in my opinion the paper can be accepted for publication.

We made all the requested structural changes and added missing citations throughout the manuscript and incorporated them to the reference list in accordance with Biology recommendations journal.

We thank reviewer#1 for his/her constructive comments.

The gene name should be written with Italic (ex ATG14). The protein name without. I suggest to make clear differentions between genes and proteins were both are discussed (ex- page 1, 41-46) EX- ATG14 gene codified the ATG protein

We apologize for not having correctly written gene and protein names throughout the manuscript. As suggested, we have now written the gene names in capitals and Italic and the protein names in capitals only.

Tian et al 2014 do not appear on reference list, Please check all the references at cite the paper according to the Biology journal recommendations.

We have now added the missing references and adjusted them following the Biology journal recommendations (i.e. ACS. Style)

Were is mentioned the paper of Tian et al 2014 is written that high expression of FIS1... So you discuss about FIS1 gene. The genes name should be written with capitals and Italic 

As mentioned above, we have now written the gene names in capitals and Italic and the protein names in capitals only.

On the page 4, row 161-163 the authors cited the paper of Nguyen et al., 2019 as a general discussion and then 163-167 I suppose that the conclusions of the mentioned study are written without references  (The protein P62 is necessary for the development and progression of AML in vivo through the induction of PINK1/PARKIN-independent mitophagy. Deletion of p62 led to the accumulation of damaged mitochondria and ROS in AML cells but had no severe impact on normal HSC, thus having no effect on hematopoiesis) Please mention the source and check this to the whole manuscript. 

We apologize for not making this clear enough in the first version of the manuscript. We have now inserted the reference of Nguyen et al. 2019 line 179 and made modifications as follow: “Another protein shown to induce poor overall survival when overexpressed is P62/SQSTM1 (Sequestosome 1), a selective autophagy receptor. Indeed, P62 protein is necessary for the development and progression of AML in vivo through the induction of PINK1/PARKIN-independent mitophagy [28]. Interestingly, P62 deletion led to the accumulation of damaged mitochondria and ROS in AML cells but had no severe impact on normal HSC, thus having no effect on hematopoiesis.’

AML has been recently characterized into specific molecular subtypes leading to a more accurate classification. Several studies have shown that, depending on the molecular subtype studied and/or the treatment used, autophagy may promote pro-tumoral or anti- tumoral effect. Please cite the paper that sustain yours information.

We are now citing the following reference: Papaemmanuil et al, NEJM 2016

Treatments have been deleted and the studies are described below.

ITD and TKD for FLT3 must be without Italic

FLT3-ITD and FLT3-TKD are now written correctly

Page 6, 183-193 several information without citations. 

We are now citing the following reference:

  • Daver et al, Leukemia 2019

FLT3-ITD expression??? 

We agree with the referee and we have changed the word “expression” to the word “activity” to be more accurate.

NPM1 (nucleophosmin 1) mutations are the most frequent genetic alteration in AML and promote aberrant cytoplasmic localization of NPM1 protein- NPM1 gene mutations .... and cite the source 

We have written the NPM1protein and NPM1 gene.

We are now citing the following reference:

  • Fallini et al, NEJM 2005

Activating mutations of the KIT receptor tyrosine kinase are also frequently detected in core-binding factor AML and are associated with higher risk of relapse. Similarly, KITD816V expression is associated with increased basal autophagy level in a STAT3 dependent manner, contributing to AML cell survival. Moreover, deletion of the key autophagy protein Atg12 strongly reduced tumor burden and improved survival of NSG-engrafted mice (Larrue et al., 2019). Genes with Italic, mention the paper cited.

After the following sentence “Activating mutations of the KIT receptor tyrosine kinase are also frequently detected in core-binding factor AML and are associated with higher risk of relapse”,

we have added the following reference:

  • Paschka et al, J Clin Oncol 2006

We have also modified the sentence “Similarly, KITD816V expression is associated with increased basal autophagy level in a STAT3 dependent manner, contributing to AML cell survival. Moreover, deletion of the key autophagy protein Atg12 strongly reduced tumor burden and improved survival of NSG-engrafted mice (Larrue et al., 2019)” to “KITD816V activity is associated with increased basal autophagy level in a STAT3-dependent manner, contributing to AML cell proliferation and cell survival in vitro. Moreover, in the same study, authors showed that deletion of the key autophagy gene ATG12 strongly reduced tumor burden and improved survival of NSG-engrafted mice (Larrue et al., 2019).”

And I can continue in this way to the end of the manuscript...

I encourage to make the structural changes of the manuscript, consisting mainly on genetics terminology (a geneticist easy can do this) and correct citations and references list, and to resubmit the paper. Once that the mentioned are corrected, in my opinion the paper can be accepted for publication.

We made all the requested structural changes and added missing citations throughout the manuscript and incorporated them to the reference list in accordance with Biology recommendations journal.

Reviewer 2 Report

The review article " Autophagy a close relative of AML biology", is a very comprehensive summary of the involvement of autophagy in AML development and treatment resistance. 

The article is very well written, with clear and interesting points of discussion. It summarises well the field and uses up-to-date literature to discuss the implications of autophagy in AML. 

 Minor comments for the article presented:

  • Page 2 line 50-51 a different font is used.
  • Page 3 line 109, journal presented in reference
  • Page 6 line 193-195, Sentence is confusing. Does inhibition of autophagy in FLT3-ITD and TKD increase the activation of other cell death pathways in AML or FLT3 inhibitor resistance is associated with autophagy directly? 
  • Following FLT3-ITD idea perhaps paragraph 4 (page 6) should come just after line 195?
  • The implications of autophagy in AML treatment is very diverse. The review presents this diversity nicely, however due to its complexity, outlined specially in pages 7 and 8, would be helpful to have an visual abstract.

Author Response

The review article " Autophagy a close relative of AML biology", is a very comprehensive summary of the involvement of autophagy in AML development and treatment resistance. 

The article is very well written, with clear and interesting points of discussion. It summarises well the field and uses up-to-date literature to discuss the implications of autophagy in AML. 

We thank reviewer#2 for his/her thoughtful comments.

Minor comments for the article presented:

Point 1: Page 2 line 50-51 a different font is used.

Accordingly changed in the text

Point 2: Page 3 line 109, journal presented in reference

Accordingly changed in the text and added in the reference list

Point 3: Page 6 line 193-195, Sentence is confusing. Does inhibition of autophagy in FLT3-ITD and TKD increase the activation of other cell death pathways in AML or FLT3 inhibitor resistance is associated with autophagy directly? 

We have changed the above sentence in the revised manuscript to “In addition, autophagy inhibition in FLT3-TKD cells, which are resistant to the FLT3 inhibitor quizartinib (AC220), is also able to inhibit proliferation both in vitro and in vivo

Point 4: Following FLT3-ITD idea perhaps paragraph 4 (page 6) should come just after line 195?

We agree with reviewer#2 and as recommended we moved the paragraph 4 just after line 217.

Point 5: The implications of autophagy in AML treatment is very diverse. The review presents this diversity nicely, however due to its complexity, outlined specially in pages 7 and 8, would be helpful to have an visual abstract.

We agree with reviewer#2 that the contribution of autophagy in AML is various and quite complex depending on AML stage, subtype or treatment. Pages 7 and 8 (8 and 9 in the new version) correspond to the paragraph 4 entitled “Autophagic response upon therapy in AML”. This part summarized in Figure 2 can represent a “visual abstract” of this paragraph. Therefore, we inserted, in the revised manuscript, Figure 2 at the end of paragraph 3, instead of to be located at the top of paragraph 3. Moreover, in the text line 275 we removed “Figure 1” in order to indicate only Figure 2.

Reviewer 3 Report

In this review by Joffre et al, they summarized the recent findings of the role of autophagy as well as mitophagy in leukemia initiation and progression. They also discussed the ambivalent role of autophagy in resistance to chemotherapies and targeted therapies.

Overall, this manuscript is of interest to both fundamental research and translational study of autophagy in the field of AML treatment. So far, numerous researches focused on targeted therapy to overcome the relapse and resistance barriers in AML. However, there are few reports elucidating the role of mitophagy as well as autophagy in AML. As mounting studies about autophagy emerging these years, this review is recommended to serve as a suggestive work to encourage more studies on autophagy and AML targeted therapy.

However, clarification and some additions as noted in the below Minor Specific Comments is warranted.

Minor Specific Comments:

  • Language of the paper needs significant improvement. The manuscript is suggested to be edited by a professional language service. The authors are advised to optimize the statements and correct those grammatical errors in this paper as well as figures (such as page 2, line 62-63; page5, line 179-180; figure 1 “pre-LCSs”).
  • As a number of studies concerning human CD34 primitive HSCs have suggested that the CD34low/- cell population contains long-term lymphohematopoietic repopulating HSCs, the “CD34- differentiated cells” described in page 2, line 83 should be corrected. (please see Sonoda, Y. (2008). Immunophenotype and functional characteristics of human primitive CD34-negative hematopoietic stem cells: the significance of the intra-bone marrow injection. J Autoimmun 30, 136-144.)
  • As there is a section of autophagy about normal hematopoietic stem cells in Part2, the subtitle of part2 is not that appropriate. It is suggested to reconsider the subtitle of part2 and part 2.2.
  • In part 4, could the authors compare and discuss the main autophagy characteristics between normal hematopoiesis and leukemia conditions? It would be of great significance to provide enlightening suggestions for autophagy-based AML target strategies.

Author Response

In this review by Joffre et al, they summarized the recent findings of the role of autophagy as well as mitophagy in leukemia initiation and progression. They also discussed the ambivalent role of autophagy in resistance to chemotherapies and targeted therapies.

Overall, this manuscript is of interest to both fundamental research and translational study of autophagy in the field of AML treatment. So far, numerous researches focused on targeted therapy to overcome the relapse and resistance barriers in AML. However, there are few reports elucidating the role of mitophagy as well as autophagy in AML. As mounting studies about autophagy emerging these years, this review is recommended to serve as a suggestive work to encourage more studies on autophagy and AML targeted therapy.

However, clarification and some additions as noted in the below Minor Specific Comments is warranted.

We thank the reviewer#3 for his/her insightful comments.

Minor Specific Comments:

Point 1: Language of the paper needs significant improvement. The manuscript is suggested to be edited by a professional language service. The authors are advised to optimize the statements and correct those grammatical errors in this paper as well as figures (such as page 2, line 62-63; page5, line 179-180; figure 1 “pre-LCSs”).

We thank the reviewer for noticing these grammatical errors. Thanks to Nathaniel Polley, a native English-speaking colleague, the sentence line 73-75 have been re-phrased to “In agreement with numerous studies, this review illustrates how autophagy's participation in AML development, as well as therapeutic resistance, transcends beyond its original function as merely a recycling process”. The sentence line 199-200 have been deleted.

In Figure 1, the typing error was corrected.

Point 2: As a number of studies concerning human CD34 primitive HSCs have suggested that the CD34low/- cell population contains long-term lymphohematopoietic repopulating HSCs, the “CD34- differentiated cells” described in page 2, line 83 should be corrected. (please see Sonoda, Y. (2008). Immunophenotype and functional characteristics of human primitive CD34-negative hematopoietic stem cells: the significance of the intra-bone marrow injection. J Autoimmun 30, 136-144.)

We agree with reviewer#3 that CD34 negative cell population contains LT repopulating HSC. In the first version of the manuscript, we wrote CD34- differentiated cells to designate CD34 negative cells but in order to be clearer, in the revised version of the manuscript, we now write CD34 negative differentiated cells. Moreover CD34+ cord blood cells have been changed to immature CD34 positive cord blood cells.

Point 3: As there is a section of autophagy about normal hematopoietic stem cells in Part2, the subtitle of part2 is not that appropriate. It is suggested to reconsider the subtitle of part2 and part 2.2.

As suggested by reviewer#3, we have now modified the subtitle of part2 and part 2.2.

Point 4: In part 4, could the authors compare and discuss the main autophagy characteristics between normal hematopoiesis and leukemia conditions? It would be of great significance to provide enlightening suggestions for autophagy-based AML target strategies.

As requested by reviewer#3, we have now incorporated these points:

- In the part 2.3 on page 6 as follows: Line 190 “In summary, normal hematopoiesis and leukemic hematopoiesis share several common characteristics with autophagy being essential to both of them. Autophagy is indeed required for maintenance and function of normal HSC (Ho et al., 2017) but also of LSC (Pei et al., 2019) mainly by regulating mitochondrial mass and hence mitochondrial functions highlighting once again the importance of mitophagy in leukemogenesis.

- In part 4 on page 10, as follows Line 391 “If the cytoprotective role of AraC-induced autophagy is confirmed, inhibiting autophagy either during AraC treatment or during the consolidation phase in clinic could represent an interesting-based AML target strategy. Moreover, since autophagy has been shown to participate to tyrosine kinase inhibitor (TKI) resistance, combination of TKI and autophagy inhibitor could also improve treatment for example in FLT3-ITD AML. Currently, chloroquine or hydroxychloroquine are the only clinically available drugs to inhibit autophagy. Unfortunately, these drugs are not specific to this process and clinical trials show that although these molecules have very little toxicity in combination with other anti-cancer treatments, its ability to modulate autophagy within the tumor remains weak (Mulcahy Levy et al., 2017, Nat Rev Cancer). The development of specific autophagy inhibitors with improved bioavailability is therefore a major challenge for the pharmaceutical industry.

However, autophagy inhibition is not a good strategy for all type of AML as it has been shown for TP53 mutated AML (see part 3). Another interesting approach is, on the other hand, to promote autophagy to eliminate oncoproteins like FLT3-ITD or PML-RAR

In conclusion, modulating autophagy in AML represent a promising therapeutic approach but need to be studied in each different molecular subgroup and each treatment approach.”
